# COVID-19 Diagnostic Strategies. Part I: Nucleic Acid-Based Technologies

**DOI:** 10.3390/bioengineering8040049

**Published:** 2021-04-17

**Authors:** Tina Shaffaf, Ebrahim Ghafar-Zadeh

**Affiliations:** 1Biologically Inspired Sensors and Actuators Laboratory (BioSA), York University, Toronto, ON M3J1P3, Canada; tshaffaf@yorku.ca; 2Faculty of Science, Department of Biology, York University, Toronto, ON M3J1P3, Canada; 3Lassonde School of Engineering, Department of Electrical Engineering and Computer Science, York University, Toronto, ON M3J1P3, Canada

**Keywords:** COVID-19 diagnostics, NA-based tests, SARS-CoV-2, point-of-care (PoC) detection, polymerase chain reaction (PCR), sequencing, microarray, CRISPR/Cas system, microfluidic-based biosensors

## Abstract

The novel Severe Acute Respiratory Syndrome Coronavirus 2 (SARS-CoV-2) has caused respiratory infection, resulting in more than two million deaths globally and hospitalizing thousands of people by March 2021. A considerable percentage of the SARS-CoV-2 positive patients are asymptomatic or pre-symptomatic carriers, facilitating the viral spread in the community by their social activities. Hence, it is critical to have access to commercialized diagnostic tests to detect the infection in the earliest stages, monitor the disease, and follow up the patients. Various technologies have been proposed to develop more promising assays and move toward the mass production of fast, reliable, cost-effective, and portable PoC diagnostic tests for COVID-19 detection. Not only COVID-19 but also many other pathogens will be able to spread and attach to human bodies in the future. These technologies enable the fast identification of high-risk individuals during future hazards to support the public in such outbreaks. This paper provides a comprehensive review of current technologies, the progress in the development of molecular diagnostic tests, and the potential strategies to facilitate innovative developments in unprecedented pandemics.

## 1. Introduction

Coronavirus Disease 2019 (COVID-19) is a respiratory infection caused by a novel coronavirus, Severe Acute Respiratory Syndrome Coronavirus 2 (SARS-CoV-2) [1]. A minimum of 2.6 million new COVID-19 patients have been reported globally in the first days of March 2021 with 63,000 new deaths [2]. It should be noted that currently, only symptomatic cases of COVID-19 are being identified and isolated, since a considerable portion of the infected cases do not experience the typical known symptoms of fever, fatigue, and dry cough [3]. Such pre-symptomatic or asymptomatic cases are not informed of being carriers and accelerate the community spread by their presence in the society, which may lead to problems such as overloaded clinics and hospitals [4].

Although vaccines such as Pfizer-BioNTech COVID-19 Vaccine have recently received approval from the U.S. Food and Drug Administration (FDA), thousands of people and many countries may have not access to them or receive the vaccines very late. Additionally, one of the current challenges is the new variants of this virus that are emerging that the authorized vaccines may not be effective against, such as the fast-spreading variant identified in the U.K. [5]. There is still a crucial need to access accurate and reliable diagnostic tests to detect not only coronavirus but also any other possible viral spreads at the earliest stages. These technologies should be flexible to be rapidly adopted for the detection of new pathogenic targets and prevent unprecedented pandemics by monitoring the disease and preventing widespread infection. Such standardized and reliable technologies will protect thousands of lives, especially in the first months of the future pandemics when no vaccines are present [6].

Generally, the nucleid acid (NA)-based tests could be performed either in a laboratory-based or in a point-of-care (PoC) setting. PoC tests include portable and miniaturized devices from benchtop-sized analyzers to small lateral flow strips; they are capable of performing tests at the same place where the samples are collected or near it. These tests have the advantage of requiring very few or no steps of sample preparation and reporting the results within minutes due to their shorter duration. These tests detect the presence of the SARS-CoV-2 virus in different facilities such as pharmacies, physician offices, and health clinics [7]. The presence and mass production of reliable, cost effective, portable, and scalable PoC tools increases the scope for easy and affordable diagnosis outside the library and in near patient or even at-home setting. By employing these techniques, it is possible to both lower the load of costly and complex diagnostic procedures and reduce time consumption during any outbreaks. The present review discusses the COVID-19 detecting technologies, their performance and challenges associated with each technology to encourage the researchers and inspire them for advancing their innovative methodologies beyond conception. In the rest of the paper, we will discuss the current lab-based and PoC strategies for NA-based detection of COVID-19. We will also highlight the potential alternative techniques that can be implemented for cost-effective, accurate, and rapid detection of infection in the future.

## 2. Nucleic Acid Amplification Tests (NAATs)

Data observed from sequencing have revealed that the new coronavirus 2019 belongs to Betacoronavirus genera with the highest pathogenicity against humans. The genetic material of this virus is a single-stranded positive-sense RNA (+ssRNA) molecule that acts as the target of COVID-19 molecular diagnostic tools [8]. Nucleic Acid Amplification Tests (NAATs) are based on amplifying the target sequence(s) followed by a read-out procedure for detection such as Fluorescent Probes, Lateral Flow Assay, DNA Sequencing, and so forth [9]. Using these techniques, SARS-CoV-2 nucleic acid can be directly targeted in the samples collected from saliva, sputum, blood, nasopharyngeal or nasal swab/wash/aspirate, throat/anal swab, bronchoalveolar lavage fluid, tissues from biopsy or autopsy, including from lung and urine [10,11]. Based on the guidelines published by the World Health Organization (WHO), NAATs are the main tests employed for the detection of the virus in suspected cases especially in the early stages of infection [11].

### 2.1. PCR-Based Tests

#### 2.1.1. RT-PCR

Reverse Transcription-PCR (RT-PCR) is the most commonly used strategy for the detection of COVID-19 in the laboratories, which has also been announced as the gold standard for testing SARS-CoV-2 infection by the WHO [11]. The combination of this technique with real-time PCR (qPCR) is frequently used to improve the technical aspect of the detection [12]. The majority of developed and commercialized NAATs for COVID-19 detection are based on conventional laboratory-based qRT-PCR technology developed to target one or multiple genes in the new viruses RNA such as ORF1ab, RdRP, E, N, and S genes [13].

Two main categories of RT-PCR are manual lab-based kits and PoC nucleic acid-based tests. In 2020, the WHO has published seven manual confirmed laboratory-based rRT-PCR protocols established by different companies or institutes including the Center for Disease Control (CDC). Dozens of companies have developed manual or semi-automated PCR-based tests that have received approval in the U.S. and other countries. These assays are the adapted version of the previous tests to target specific sequences in the novel virus genome. The tests may vary in the target gene(s), sensitivity, specificity, the limit of detection (LoD), and total turn-around time (TaT) [13]. The sensitivity and specificity of the RT-PCR-based tests are often comparable; however, some of them have the advantage of accepting more sample types due to their higher flexibility. As an example, mid-turbinate swabs are accepted by a few assays including Thermo Fisher Scientific’s test [14]. Another important characteristic is the throughput of the instrument compatible with the test; performing the test for a higher number of the samples simultaneously is merit for some of the tests such as PerkinElmer^®^ SARS-CoV-2 real-time RT-PCR Assay and the TaqPath COVID-19 Combo Kit, which perform up to 96 samples while the Alinity SARS-CoV-2 assay requires 2–3 h to report the results for only 24 samples. Some of the FDA Emergency Use Authorization (EUA) approved manual qRT-PCR kits are explained and compared in Table 1. Although these kits benefit from high sensitivity and specificity, they require multiple hands-on steps and a long hands-on time (HoT) to report the final result. Hence, they are suitable for a lab-based early detection setting but not for rapid and near-patient diagnostics [15].

On the other hand, fully automated or sample-to-answer assays are instrument-based tests performing all of the steps automatically in a mobile- or in a facility-based platform that makes them suitable for PoC detection of infection. Compared with manual kits, they are operated faster by less technical staff and technical training, do not need infrastructural requirements, requiring only a few minutes of HoT, which frees up the technicians and observing less contamination in the samples due to the fully automated system [16]. Some of the well-known companies have developed their previous sample-to-answer technologies to sense novel coronavirus 2019, and their assays are commonly being used such as Roche Cobas^®^SARS-CoV-2 assay, Xpert Xpress SARS-CoV-2 test, GenMark ePlex SARS-CoV-2 test, NeuMoDx™ SARS-CoV-2 assay, Abbott RealTime SARS-CoV-2 assay, and Hologic Aptima^®^ SARS-CoV-2 assay. Roche Cobas^®^ SARS-CoV-2 test is a laboratory-based kit performing 96 tests in ≈3 h. This assay targets the ORF1 ab non-structural region and a conserved part of the E-gene on SARS-CoV-2 RNA. An LoD of ≤ 10 copies/mL is achieved for this test using clinical samples. However, this test is not suitable for PoC detection due to some limitations such as a huge instrument and requirement for trained technicians [17,18]. A Cepheid Xpert^®^ Xpress SARS-CoV-2 rapid and qualitative test is the first automated assay authorized to be used in PoC setting, but it needs to be performed by trained technicians. This assay targets E, RdRp, ORF1a, and N gene sequences in RNA viruses [19]. Compared with the Cobas^®^ SARS-CoV-2 test, this system is much smaller with a lower throughput of 1-80 modules, it has the advantage of requiring shorter HoT and reporting the results within 45 min; the specificity and the sensitivity of this assay equal are 97.8% and 95.6% respectively with an LoD of 0.0200 PFU/mL [20]. Moran and colleagues evaluated a Roche Cobas SARS-CoV-2 Assay with Cepheid Xpert Xpress SARS-CoV-2 tests and based on the evidence, high-throughput laboratory-based assays achieve lower LoDs and higher sensitivities compared with rapid mobile analyzers [21]. Xpert^®^ Xpress SARS-CoV-2/Flu/RSV is another kit that is recently approved to track six respiratory viruses with an LoD of 131 copies/mL for SARS-CoV-2 [22].

QIAstat-Dx Respiratory SARS-CoV-2 Panel is a rapid PoC test that has the advantage of requiring no PCR-trained laboratory technicians [23]. This respiratory panel has the advantage of simultaneous detection of 21 different pathogens including SARS-CoV-2 by targeting ORF1b and the E genes. Evaluation of this kit in comparison with the WHO-PCR workflow has demonstrated a sensitivity of 100% and a specificity of 93% with no cross-reaction [24]. The ePlex SARS-CoV-2 Test is another automated lab-based or PoC test that targets only one sequence, SARS-CoV-2 N gene [25]. The throughput of the system varies from 3 to 28 samples. The literature has reported an LoD of 1000 copies/mL for the ePlex test, which is considerably lower than the LoD stated by the company while submitting for EUA (100,000 copies/mL based on in vitro tests) [26]. As limitations of the ePlex test, the specificity of the test is decreased at high titers and cross-reactivity is observed with SARS CoV-1 [27].

The GenomEra SARS-CoV-2 Test is a rapid multiplex RT-PCR assay targeting its RdRp and E genes [28]. The throughput of the system is lower compared with the other automatic tests, 1–4 samples, the HoT is about 5–10 min for four samples, and the results are reported in 70 min. The benefit of this test is detecting pathogens without requiring the RNA-extraction step, which significantly reduces the technicians’ workload and TAT [29]. The ANDiS^®^ SARS-CoV-2 RT-qPCR Detection Kit amplifies ORF1ab, N, and E genes using the RT-qPCR method. In this kit, primers and probes simultaneously target SARS-CoV-2, Flu A and Flu B virus-specific. The kit is highly sensitive with an LoD of 5 copies/reaction, its performances were compared with NGS, and the results demonstrated a sensitivity of 96% for the SARS-CoV-2 RT-qPCR Detection Kit with a specificity of 100%. The final results of coronavirus detection will be available after a TaT of 60 min PCR program [30].

The ARIES^®^ SARS-CoV-2 Assay is by Luminex Company (Austin, TX, USA), its HoT is about two minutes, the throughput is up to 12 samples, and the results are delivered in about two hours. This assay uses two ORF ab and N genes to detect SARS-CoV-2 [31]. The test has both sensitivity and specificity of 100%, these statistics require to be confirmed using clinical experiments though [32]. BD MAX™ automated system performing qRT-PCR Reagents including primers and probes based on CDC protocol requiring trained laboratory technicians. The test targets are targets N1 and N2 regions in the Nucleocapsid gene of SARS-CoV-2 genome with about a one-minute HoT per sample, a 15-min HoT per run, and a TaT of about three hours for up to 12 samples [33]. One of the concerns associated with this assay is its false-positive results. In July 2020, the FDA warned the clinical laboratory staff and health care providers of a high risk of about 3% for obtaining a false-positive result using BD SARS-CoV-2 Reagents for the BD Max System test and mentioned that the results of this test should be confirmed using a second authorized test [34].

**Table 1 bioengineering-08-00049-t001:** A selected list of the FDA EUA approved real-time PCR-based tests and their performance for COVID-19 detection.

Company	Test Name	Target Gene(s)	Sensitivity	LoD	Specificity	Assay Time
Vela Diagnostics (Singapore) [35]	ViroKey™ SARS-CoV-2 RT-PCR Test	ORF1a, RdRp	97.2%	(ORF1a: 250 genome equivalents (GE)/mL, RdRp: 560 GE/mL	95.1%	3.5 h
Verily Life Sciences (San Francisco, CA, USA) [36]	Verily COVID-19 RT-PCR Test	ORF1ab, N gene, S	100%	60 GE/mL	100%	(No info)
MiraDx (Los Angeles, CA, USA) [37]	MiraDx SARS-CoV-2 RT-PCR assay	N1, N2	96.90%	4000 copies/mL	100%	2–4 h
BayCare Laboratories, LLC (Tampa, FL, USA) [38]	BayCare SARS-CoV-2 RT PCR Assay	ORF1, E gene	88%	0.009 TCID50/mL	100%	(No info)
DxTerity Diagnostics, Inc. (Rancho Dominguez, CA, USA) [39]	DxTerity SARS-CoV-2 RT PCR CE Test	N gene, E gene, ORF1ab	97.3%	50 copies/mL	90.0%	2–4 h
Texas Department of State Health Services, Laboratory Services Section (Austin, TX, USA) [40]	Texas Department of State Health Services (DSHS) SARS-CoV-2 Assay	N gene and ORF1ab	100.0%	20 copies/mL	100%	(No info)
Yale School of Public Health, Department of Epidemiology of Microbial Diseases (New Haven, CT, USA) [41]	SalivaDirect	N gene (N1 region)	94.1%	6000 copies/mL	90.9%	≈2 h
Solaris Diagnostics (Nicholasville, KY) [42]	Solaris Multiplex SARS-CoV-2 Assay	N gene (N1 and N2 regions)	100%	10,000 copies/mL	100%	2–4 h
Alpha Genomix Laboratories (Peachtree Corners, GA, USA) [43]	Alpha Genomix TaqPath SARS-CoV-2 Combo Assay	ORF1ab, N, S	96.70%	4000 copies/mL	100%	2–4 h
George Washington University Public Health Laboratory (Washington, DC, USA) [44]	GWU SARS-CoV-2 RT-PCR Test	N gene (N1 and N2 regions)	95.00%	12,500 copies/mL	100%	(No info)
Wren Laboratories (Branford, CT, USA) [45]	Wren Laboratories COVID-19 PCR Test	N1 of SARS-CoV-2, N3 of Sarbecovirus	100%	10,000 copies/mL	95.0%	(No info)
Ethos Laboratories (Newport, KY, USA) [46]	Ethos Laboratories SARS-CoV-2 MALDI-TOF Assay	Orf1ab, N1, N2, N3, ORF1	98.10%	1 TCID50/mL	96.3%	2–4 h
Cleveland Clinic Robert J. Tomsich Pathology and Laboratory Medicine Institute (Cleveland, OH, USA) [47]	Cleveland Clinic SARS-CoV-2 Assay	E, RDRp	97.0%	10,000 copies/mL	100%	(No info)
ISPM Labs, LLC dba Capstone Healthcare (Atlanta, GA, USA) [48]	Genus SARS-CoV-2 Assay	N (2 targets)	100.0%	40,000 copies/mL	100%	(No info)
Abbott Molecular Inc. (Des Plaines, IL, USA) [49]	Alinity m SARS-CoV-2 assay	RdRp, N	100%	100 copies/mL	100	<115 min to 12 first results, 16 min thereafter
altona Diagnostics (Hamburg, Germany) [50]	RealStar SARS-CoV-2 RT-PCR Kit U.S.	E, S	No info.	1.00 E-01 PFU/ml	100%	4–6 h
Beijing Wantai Biological Pharmacy Enterprise Co. Ltd (Beijing, China) [51]	Wantai SARS-CoV-2 RT-PCR Kit	ORF1ab, N	100%	50 copies/mL	100%	2–4 h
bioMérieux SA (Marcy-Letolle, France)	SARS-COV-2 R-GENE^®^	N, E, RdRP	100%	380 copies/mL	100%	<1 h
EUROIMMUN AG (Lubeck, Germany) [52]	EURORealTime SARS-CoV-2	ORF1ab, N	100%	1 copy/µl	100%	
Sansure Biotech Inc. (Changsha, China) [53]	Novel Coronavirus (2019-nCoV) Nucleic Acid Diagnostic Kit (PCR-Fluorescence Probing)	ORF1, N	94%	200 copies /mL	99%	1 h, 15 min
SD Biosensor Inc. (Suwon-si, Korea) [54]	STANDARD M nCoV Real-Time Detection Kit	E, ORF1ab	No info.	0.5 copies /µL for upper respiratory specimens and 0.25 cp/µL for lower respiratory specimens	100%	6 h
Seegene Inc. (Seoul, Korea) [55]	Allplex™ 2019-nCoV Assay	E, N, RdRP	No info.	4167 copies/mL	100%	1 h, 50 min
Thermo Fisher Scientific (Waltham, MA, USA) [56]	TaqPath™ COVID-19 CE-IVD RT-PCR Kit	ORF1ab, S, N	100%	1250 copies/mL	97%	4 h

Spartan Biosciences Inc. (Ottawa, ON, Canada) in Ottawa has developed a laboratory-in-a-box technology and received Health Canada approval for its Spartan Cube Covid-19 System to be performed in hospitals or by healthcare professionals using the Spartan Cube system, which is the smallest DNA analyzer in the world and has recently received approval from Health Canada on 22 January 2021 [57]. NeuMoDx™ SARS-CoV-2 Assay [58] and AIGS assay [59] as two other PoC molecular tests and DiaSorin Molecular Simplexa™ COVID-19 Direct qRT-PCR assay [60] as a fully automated lab-based test are some other assays developed for PoC COVID-19 detection.

Even the fastest and the most accurate RT-PCR-based tests suffer from some shortcomings. There are mostly high costs for purchasing both the instrument and the required materials, especially for high-throughput tests and for this reason, a vast majority of people do not have access to the equipped centers for such tests globally. While using the automated system, all of the inserted samples are consumed inside the instrument, and there is no access to the extracted nucleic acid to perform additional tests, especially when the result is negative and supplementary tests are required. Additionally, these systems are quantitative and do not detect the viral load in the sample for more specific analysis [27,61].

#### 2.1.2. ddPCR

Droplet Digital PCR (ddPCR) is one of the potential strategies to increase the sensitivity and accuracy of the conventional PCR tests by reducing the false-negative results, especially when the patient is a weak positive with a low virus load [62]. Bio-Rad laboratories in a partnership with Biodesix Inc. have developed a Droplet Digital™ PCR (ddPCR™) technology for COVID-19 detection [63]. The Bio-Rad SARS-CoV-2 ddPCR™ Kit is an FDA EUA approved qualitative partition-based endpoint RT-PCR test targeting N1 and N2 regions in the viral N gene with 100% accuracy and an LoD of 625 copies/mL, which means that this test is capable of detecting the viral genetic materials when there are 25 or more copies of them in 1 mL sample. Additionally, it demonstrated no cross-reaction with other pathogens [63]. The main disadvantage is the long duration, since the results are reported 24–48 h after sample collection, which has limited applying ddPCR in medical laboratories, the duration of ddPCR workflow is approximately 2 h higher than qRT-PCR (15% more time), and the average costs for the mandatory equipment and consumables for ddPCR is about 5–10% higher than qRT-PCR [64,65]. Distinct clinical evaluations have obtained high accuracy as well as low LoD capable of detecting 2 viral copies/mL. The limit of detection is considerably low for this test in comparison with most of the other kits, and it can detect the copy numbers of the virus RNA as low as 2 copies in in one milliliter of the introduced sample [66]. This strategy has the potential of increasing the signal-to-noise ratio and requires lower amounts of materials, which results in making it more cost-effective [67]. A comparison between the performance of RT-qPCR with ddPCR strategy has illustrated that qRT-PCR is not capable of detecting low viral loads owing to its insufficient sensitivity. Another important advantage of ddPCR over qRT-PCR is that using micro dilutions results in fewer effects interference of any reaction inhibitors on the system, and the results are more repeatable and robust [65]. It can be concluded that ddPCR improves the diagnostic procedure for the early detection of COVID-19 and also the effect of the therapeutic interventions and drug doses with high accuracy, but this strategy is not suitable for rapid PoC detection of this disease [68].

#### 2.1.3. nPCR

Nested PCR (nPCR) is an amplification-based technology employing more than one forward and reverse oligonucleotide primer set [69]. The sensitivity of the standard RT-PCR can be improved by applying an additional nested PCR on the primary amplicons [70]. The BioFire^®^ COVID-19 Test is one of the sample-to-answer automated assays taking advantage of nPCR technology for COVID-19 detection (Figure 1) [71]. This nested multiplexed real-time RT-PCR test targets ORF1ab and ORF8 sequences and reports qualitative results in ≈50 min [71] with 90% sensitivity, 100% specificity, and an LoD equal to 330 copies/mL [72]. Another BioFire^®^ authorized qualitative multiplex test is BioFire^®^ Respiratory 2.1 (RP2.1) Panel detecting SARS-CoV-2 and 21 additional respiratory viruses and bacteria simultaneously, a high-throughput and automated assay with ≈2 min HoT. The sensitivity and the specificity of the RP2.1 Panel are 97.1% and 99.3%, respectively, and the estimated LoDs related to the panel for SARS-CoV-2 detection is 6.9E-02 TCID50/mL for heat-inactivated virus and 1.6 × 10^2^ copies/mL for the infectious virus [73]. Clinical evaluation of this panel yielded a slightly higher accuracy with 98% sensitivity and 100% specificity. These studies have demonstrated that this panel along with four other assays including BioFire^®^ Defense COVID19, Roche Cobas, and Cepheid Xpert Xpress were capable of detecting lower viral loads compared with ID NOW and Hologic Aptima tests (explained in the next sections), making them reliable tools even in the acute presentation phase of the infection low viral titer detection [74].

SARS-CoV-2 TEM-PCRTM Test is another FDA EUA authorized assay developed by Diatherix Eurofins Laboratory. The technology underlying this qualitative assay is nested end-point PCR followed by hybridization [75]. Diatherix Eurofins Scientific takes advantage of a private technology, TEM-PCR™ (Target Enriched Multiplex PCR), which is based on target-specific nested primers. The main characteristic of this technology is the enrichment of different targets by using primer mixes. Clinical evaluations for this assay have reported an LoD of as low as 1 copy/µL; the sensitivity and specificity of the test are estimated to be 100% and 98% respectively, and no false-positive was obtained [75]. The main weakness of this technology is an increased risk of contamination due to the sequence of manual steps, and this is the reason why this technique is not usually the preferred diagnostic test to be performed in clinical laboratories. Additionally, the necessity of performing two manual steps of nPCR elevates the workload and HoT, which makes it not applicable as a PoC test [76,77].

### 2.2. Isothermal Nucleic Acid Amplification-Based Tests

Due to the current limitations of PCR-based strategies, the development of other potential diagnostic strategies has drawn great interest globally. Isothermal nucleic acid amplification is an alternative strategy with a short TaT allowing amplification at a constant temperature, and it eliminates the need for a thermal cycler [78,79]. Such advantages are the reasons now some detection methods are under development or have received approval based on this principle. This category includes the promising approaches with the potential of being used in combination with other systems as the amplification step to address the challenges [80,81].

#### 2.2.1. LAMP

Loop mediated isothermal amplification (LAMP) is one of the alternative strategies for the quantitative detection of respiratory viral infection. The test is initiated by either inserting RNA samples and performing Reverse Transcription (RT)-LAMP, or inserting cDNA samples for LAMP amplification [82]. This technology employs four to six or even eight primers to sensitively identify different regions within the target sequence [83]. LAMP is suitable and ready for rapid lab scale-up and high-throughput automation for pathogenic detection, and some researchers have already used this strategy for COVID-19 as well (Table 2). The Lucira COVID-19 All-In-One Test Kit is a RT-LAMP-based assay, the only prescription home testing kit given FDA EUA to be used in PoC, and also at a home setting for suspected people older than 14 [84]. Such kits may act as game-changers in the near future as rapid PoC at-home tests; however, it is necessary to employ them in a non-clinical setting for more supporting research and collecting data, hence, the current information is not reflecting the use of the kits in the real world. The other potential drawbacks are sample-type limitation, being operator dependent and vague performance characteristics for at-home usage [85].

The Abbott Diagnostics automated test, namely ID NOW COVID-19 assay, is one of the first PoC NA-based tests. This rapid and qualitative instrument-based test targets RdRp gene and provide positive results in as low as ≈5 min and negative results in 13 min [86]. The sensitivity of this assay is measured at 71.7% with an LoD of 125 copies/mL [86,87]. This test is reported to achieve a high number of false-negative results, especially while performing tests in the two first weeks of the infection [88], and it has also been criticized for its false-negative rate [89]. The rate of the false-negative detections can be related to the sample type and/or its low viral load; considering the LoD of the test, weak positives are not accurately detected in the early stages of the infection [90].

Gun-Soo Park et al. used an RT-LAMP assay for the detection of SARS-CoV-2 with an LoD 100 copies/reaction in 30 min, but this test was not low enough for early detection of the infection as well. This problem could be resulted from inappropriate target selection and might be improved by choosing more appropriate target sequences for LAMP amplification [82]. Yan and colleagues have tried to increase the sensitivity of the LAMP test by targeting reliable targets in ORF1 ab and S genes in separated tubes with five and six primer sets respectively, the results were achieved in ≈26 min with an LOD of 20 copies/reaction and no cross-reaction with other respiratory pathogens, while the sensitivity and specificity of the test were 100% for 130 clinical cases [91].

Compared with RT-PCR, LAMP is faster, and it targets multiple sequences within the target DNA with no expensive thermocyclers without facing supply chain issues. It also amplifies longer sequences with a comparable sensitivity with PCR, produces much higher amounts of DNA compared with qRT-PCR, and its visual interpretation has made it independent of any costly instruments [92]. To improve the sensitivity, this technology is widely used in combination with other sensing strategies such as CRISPR-based systems as a pre-amplification step for boosting the accuracy of the test [93,94,95]. Due to its limited background in the literature compared with RT-PCR, LAMP technology is in the position of being assessed and improved in a clinical setting for COVID-19 detection [92].

#### 2.2.2. RPA

RPA (Recombinase polymerase amplification) is another isothermal amplification strategy used for the detection of viral nucleic acid. The advantage of the high sensitivity of RPA results from adding extra probes to the test [96]. RPA-based viral detecting assays are able to detect low concentrations of pathogenic RNA or DNA faster than PCR or other isothermal amplification techniques [97,98]. This technology is known as a highly sensitive, specific, cost-effective, simple, and compatible amplification method. The test is very fast, requires lower temperature compared with LAMP, and there is no need for adding multiple primers or denaturation at the first stage. These characteristics along with amplifying low nucleic acid concentrations make this equipment-free technique suitable to be performed in a wide range of assays in PoC settings [99]. However, the necessity of manual intervention increases the HoT time and probability of contamination. Additionally, various results may be yielded in a user-dependent manner, and similar to the other tests, the performance of the RPA reagents could possibly change based on the storage circumstances [100,101].

RPA has been used in combination with other technologies such as the lateral flow immunoassay (LFIA) system for diagnostic purposes [102]. It has proven to be a potential strategy for pre-amplification of the target sequences prior to CRISPR-based nucleic acid detection to provide the required products for CRISPR/Cas biosensing systems, which are explained in more detail in the next section [103].

**Table 2 bioengineering-08-00049-t002:** A selected list of the developed LAMP-based tests and their performance for COVID-19 detection.

Manufacture	Test	LoD	Sensitivity	Specificity	Target	Duration	Regulatory Status
Lucira Health, Inc. (Emeryville, CA, USA) [104]	Lucira COVID-19 All-In-One Test Kit	900 copies/mL	A single nucleotide mismatch is probable in one of the primers (Positive agreement: 94%, Negative agreement: 98%)	No cross-reaction	N	30 min	FDA EUA
Detectachem Inc. (Sugar Land, TX, USA) [105]	MobileDetect Bio BCC19 (MD-Bio BCC19) Test Kit	30% for 25 copies/mL and 100% for 75 copies/mL	Positive agreement: 97.7% Negative agreement: 100%	Cross-reaction with SARS-CoV	N, E	30 min	FDA EUA
SEASUN BIOMATERIALS, Inc. (Seoul, Korea) [106]	AQ-TOP COVID-19 Rapid Detection Kit PLUS	1 copy/µL	(No info)	Some primers have homology with other microorganisms	ORF1ab, N	20 min	FDA EUA
UCSF Health Clinical Laboratories, UCSF Clinical Labs at China Basin (San Francisco, CA, USA) [107]	RT-LAMP	20,000 copies/mL	95%	100.0%	N (N2 region)	45 min	FDA EUA
Abbott Diagnostics Scarborough, Inc. (Scarborough, ME, USA) [108]	ID NOW COVID-19	125 copies/mL	71.7%	no cross-reactivity	RdRp	Positive results 15 min, Negative results 30 min	FDA EUA
Pro-Lab Diagnostics (Round Rock, TX, USA) [109]	Pro-AmpRT SARS-CoV-2 Test	125 genomic equivalents/swab	96.60%	100.0%	ORF1ab	30 min	FDA EUA
Color Genomics, Inc. (Burlingame, CA, USA) [110]	Color SARS Cov-2 Diagnostic Assay	0.75 copies/μl	100%	100%	ORF1a, E, N,	70 min	FDA EUA
SEASUN BIOMATERIALS [106]	AQ-TOP™ COVID-19 Rapid Detection Kit	7000 copie/ml	(No info)	(No info)	ORF1ab	Positive results 15 min, Negative results 30 min	FDA EUA
Atila BioSystems Inc. (Mountain View, CA, USA) [111]	Atila iAMP^®^ COVID Detection Kit	~2000 copies of viral RNA per swab	100%	99%	ORF1ab, N	75 to 90 min	FDA EUA
CapitalBio Technology (Beijing, China) [112]	Respiratory Virus Nucleic Acid Detection Kit	5 × 10^2^ copies per reaction	(No info)	(No info)	(No info)	13 for respiratory pathogens simultaneously	CE-IVD

CE-IVD: approved CE Marking according to be sold in Europe.

#### 2.2.3. TMA

Transcription-Mediated Amplification (TMA) amplifies the target sequences much more efficiently compared with RT-PCR-based assays without requiring a thermal cycler. The high rate of sensitivity can be related to either the extraction step or amplification step or both [113]. TMA amplification has an autocatalytic nature which is expected to efficiently generate more RNA amplicons than PCR-based assays [114]. The products of these technologies are detectable using colorimetric assay, fluorescent probes, and gel electrophoresis [115].

Hologic, Inc. is one of the centers focusing on TMA strategy for detection. After the initiation of the COVID-19 outbreak, they have developed their proprietary technology for SARS-CoV-2 detection called Aptima SARS-CoV-2 assay [116]. This fully automated test is one of the commonly used tests, which amplifies two conserved sequences in ORF1ab gene. The analytical sensitivity and specificity of this test are measured 0.026 TCID_50_/mL and 100%, respectively. Fully automated systems are used to perform this test, which decreases the HoT time and contamination probability; however, it requires trained technicians, and qualitative results are reported in up to 3.5 h [117]. Studies have demonstrated comparable or even higher analytical sensitivity for COVID-19 detection using Hologic Aptima SARS-CoV-2 compared with RT-qPCR due to higher sensitivity of TMA, which makes it helpful for high-throughput and rapid detection of infection in laboratories [118,119].

### 2.3. CRISPR/Cas-Based Tests

Clustered regularly interspaced short palindromic repeats/CRISPR-associated (CRISPR/Cas) systems are powerful and specific tools for biosensing nucleic acids and genome editing in different fields [120]. One of the promising applications for this system is DNA or RNA-targeting in which the CRISPR/Cas system is used for signal generation in the detection step in combination with an additional pre-amplification step [121]. The prior pre-amplification procedure is mostly an isothermal amplification such as LAMP and RPA, which increases the sensitivity of the assay by amplifying the target sequences and decreasing the LoD. The final step is signal reporting, the readout of these fluorescent signals is performed using agarose gel, quenched fluorescent, or visual detection when integrated with lateral flow assays (LFA) [121,122,123].

The COVID-19 detecting assays that employ the CRISPR/Cas system are mainly all rapid, portable, sample tolerant, highly accurate even for the detection of single-base variations, simple to develop or redevelop, independent of any expensive instruments or traditional infrastructures necessary in traditional molecular laboratories, and have extremely low costs per sample [124]. These tests require a combination of materials different from PCR, offering a potential alternative during chemical shortages. Another merit of this technology is the capability of being combined with a paper strip to detect the presence of SARS-CoV-2. Considering all of the aspects, CRISPR/Cas systems are novel and improving technologies suitable for large-scale screening in near-patient settings. However, they suffer from integrated sample preparation and its complications, limited target regions and issues for multiplexed sensing [125]. It should be noted that their applications are limited to lab-based diagnostic tests due to a series of manual steps of mixing and incubation. The necessity of an additional amplification step and sample pretreatment increase the workload and total HoT from a few minutes to hours, which is the other demerit to be eliminated in the future [126,127].

The pioneer scientists in this field, Sherlock Biosciences and Mammoth Biosciences, along with other manufactures have developed their patented methodology for SARS-CoV-2 detection (Table 3) [128]. Each kit contains a specific Cas protein, isothermal amplification procedure, and monitoring technology. SHERLOCK (specific high sensitivity enzymatic reporter unlocking) is the first CRISPR-Cas13 platform designed in combination with RT-RPA [103]; SHERLOCKv2 is the revised platform of SHERLOCK, which is capable of detecting more than one target sequence simultaneously [129]. SHERLOCK COVID-19 detection protocol is completed in 1 h followed by a paper-based visual readout (Figure 2). The assay is capable of detecting SARS-CoV-2 RNAs with an LoD of 10–100 copies/µL [130]. However, this lab-based method is complicated; it includes two distinct steps of reaction with an increased probability of cross-contamination due to requiring sequential steps of manual fluid handling and opening the tubes. Hence, researchers have used the STOP (SHERLOCK Testing in One Pot) strategy and simplified SHERLOCK assay to make it suitable for COVID-19 PoC testing outside the library. In this assay, the RPA reaction is replaced with LAMP amplification targeting the N gene. The STOPCovid test turns out a result in 40 min when using fluorescence readout and 70 min when with LF readout. The obtained LoD for this test is 100 copies/reaction for the strip-based setting [131]. STOPCovid.v2 is the new version of the STOPCovid test adapted with a magnetic bead purification step for increasing sensitivity. This assay achieved positive results in 15–45 min with 93.1% sensitivity and 98.5% specificity. Although this simplified format requires further development, it is suitable to be performed in a PoC setting [132].

DETECTR (DNA endonuclease-targeted CRISPR trans reporter) is developed by Mammoth Biosciences and GSK to target the SARSS-CoV-2 N and E genes with the high sensitivity using the CRISPR-Cas12 system and visual lateral flow strip [133]. The duration of the test is 30–40 min with an LoD of 70–300 copies/µL. This test is much faster compared with SHERLOCK COVID-19 assay, but its sensitivity is lower, especially for early detection of the disease, which may result in more false-negative results; the evaluations have also demonstrated cross-reactions with SARS-like coronaviruses [134].

Many other research groups have focused on CRISPR-based solutions for the detection or other purposes to fight COVID-19 [135,136,137]. Abbot et al. developed a CRISPR-Cas13-based method termed PAC-MAN (prophylactic antiviral CRISPR in human cells) as an antiviral strategy for degrading RNA from Influenza A and SARS-CoV-2 viruses in human respiratory epithelial cells and successfully decreased the viral load in these cells by targeting at least 90% of the viral particles and showed to be a potential inhibitory technology [138]. Curti and colleagues evaluated a CRISPR-Cas12 based diagnostic tool with RT-RPA for SARS-CoV-2 sensing and reported a LoD of 10 copies/μL for ORF1ab [128]. One other proposed assay for SARS-CoV-2 and HIV qualitative detection is an all-in-one dual CRISPR-Cas12a (AIOD-CRISPR) assay, which is validated using COVID-19 patients’ samples. The duration of the test is just a few minutes with a sensitivity of 4.6 copies. After a 40-min incubation at 37 °C, 1.3 copies of DNA targets were detected with no cross-reaction with other tested respiratory viruses [139].

**Table 3 bioengineering-08-00049-t003:** CRISPR-based tests for COVID-19 detection.

Manufacture	Test	Technique	Detection	Test Format	Target	Time to Result	Analytical Sensitivity (LoD)	Specificity	Regulatory Status
Sherlock Bioscience [131]	Sherlock™ CRISPR SARS-CoV-2	RT-LAMP + CRISPR/Cas	Lateral-flow visual readout	Rapid PoC test	ORF1ab, N	1 h	6.75 copies/uL	No cross-reaction	FDA EUA
Mammoth Biosciences [134]	SARS-CoV-2 DETECTR Reagent Kit	RT-LAMP + CRISPR/Cas12	Lateral-flow visual readout	PoC High-throughput diagnostic	N and E	30–40 min	20–30 copies/µL	No cross-reaction	FDA EUA
Caspr Biotech [140]	Lyo-CRISPR SARS-CoV-2 Kit	RT-LAMP + CRISPR-Cas12	Fluorescence detection using reader	Semi-automated, High throughput (48 tests), using Lyophilized beads	Direct from Sample Kit: 2 regions in N and 1 region in orf1ab; Purified RNA kit: 1 region in N	≈1 h	Direct from Sample Kit: 25 copies/μL; Purified RNA kit: 7.5 copies/μl	100%	In review for FDA EUA

### 2.4. DNA-Microarray Based Tests

Microarray is a valuable technique for quantitative detection and genotyping the viral nucleic acid. Microarrays consist of thousands of DNA oligonucleotides as probes able to identify different nucleic acids simultaneously exhibiting a significantly higher specificity and sensitivity in comparison with the tests targeting only one sequence [141]. These tests are not commonly used for PoC purposes, and their main application is achieving information about gene expression levels, genotyping, characterizing the DNA or RNA for the detection of mutations, and some other novel applications [142].

After the initiation of the COVID-19 pandemic, PathogenDx’s novel DetectX-RV combined multiplex end-point RT-PCR with DNA microarray to improve the specificity of the test and create new possibilities for multiplex testing. This multiplex assay contains five primer sets targeting SARS-CoV-2 N1, N2, N3 genes, and the SARS-CoV N2 gene. The results are ready in 6–8 h. Although this test has not received FDA EUA yet, it employs up to 12 specific probes, which is a higher than most of the authorized tests with a throughput of 96 tests per kit. The TaT for microarray hybridization is one hour, and the LoD of the test is observed between 50 and 250 copies/reaction for three low to high viral concentrations. However, similar to the other nucleic acid-based tests, microarrays may yield false results in various circumstances [143,144].

Alimetrix, Inc. (Huntsville, AL, USA) has developed the FDA EUA authorized Alimetrix SARS-CoV-2 RT-PCR Assay for COVID-19 detection. This assay combines RT-PCR and microarray technologies to target SAES-CoV-2 ORF1ab, N1, and N2 genes [145]. VereRTCoV™ SARS-CoV-2 Real-Time RT-PCR 2.0 Kit is another qualitative microarray-based test that combines microfluidics, molecular biology, and microelectronics. In June, this test has received CE-IVD marking to be used in the clinical setting in Europe [146]. This test targets two sequences in the viral N gene and human RNase P gene including a long HoT and many manual steps making it unsuitable for PoC detection as a rapid near-patient test. The duration of the run is ≈1.5 h with a sensitivity of two RNA copies/reaction and no cross-reactivity with any human or other respiratory nucleic acid [147].

Lumex Instruments Canada is another company producing an RUO Microchip RT-PCR COVID-19 detection system targeting N1 and N2 primer-probes target sequences in the viral N gene and human RNase P control gene. The results are achieved in 50 min, it requires low amounts of reagents, the LoD of the assay is 9 × 10^3^ copies/mL, and the costs are lower compared with PCR [148,149]. Genomica SauÂ is a Spanish company that has developed a CLART^®^COVID-19 test based on their patented CLART^®^ technology. In this assay, a multiplex-PCR amplifies the targets, which are followed by inserting the products into a low-density microarray and hybridization with specific probes. The results are reported within five hours with 96% specificity and 98% sensitivity, and the throughput is up to 96 samples [150].

When the aim of the research is investigating a few genes or mutations, microarrays compete with PCR-based tests. On the other hand, complex microarrays are capable of performing a large number of tests and provide huge information while PCR-based tests are not; in such cases, the competition is between microarray and next-generation sequencing (NGS) technique [151]. The main challenges of designing microarray-based tests are the presence of highly conserved sequences in coronaviruses RNA as well as the incidence of cross-reaction between coronaviruses genomes [152]. Although the costs are usually high for microarray-based tests, low-cost microarrays have also been developed to investigate coronavirus strains with a sensitivity comparable to qRT-PCR though [153].

## 3. Sequencing-Based Tests

Next-generation sequencing (NGS) is a high-throughput method for studying the whole genomes, some parts of the genetic material, or the transcripts in the cells [154]. Three main strategies have been employed for SARS-CoV-2 RNA sequencing: whole-genome sequencing, direct RNA sequencing, and metagenomic sequencing. On 10 January 2020, researchers uploaded the full genetic sequence of SARS-CoV-2 from positive SARS-CoV-2 samples to the GISAID platform, and the sequences of thousands of SARS-CoV-2 genomes have been uploaded to this database [155]. GenBank and the Sequence Read Archive of the US National Center for Biotechnology websites also have several full-length sequences of SARS-CoV-2 from different regions [156,157]. Random-amplification deep-sequencing approaches have been playing a crucial role in identifying and direct investigating infectious MERS-CoV and SARS-CoV-2 viruses. Such deep-sequencing strategies such as next-generation sequencing (NGS) and metagenomic next-generation sequencing (mNGS) will remain beneficial for the determination of future mutations and variants of SARS-CoV-2 [158].

FDA has authorized the first NGS test, Illumina (San Diego, CA, USA) COVIDSeq Test, for coronavirus 2019 detection in June 2020 in order to generate information on viral genetic material, monitoring the mutations and finding the reason for the genetic variations, which is critical for fighting the virus [159]. Illumina has developed a Shotgun metagenomic sequencing strategy using illumine sequencing systems for the qualitative detection of novel SARS-CoV-2. This assay employs 98 amplicons to amplify specific sequences on SARS-CoV-2 RNA. The test is capable of performing 302–384 tests within 12 h based on the instrument [160]. Thermo Fisher is another manufacturer that has announced an Ion AmpliSeq SARS-CoV-2 test for COVID-19 identification, sequencing, surveillance, and epidemiology research. This panel covers more than 99% of SARS-CoV-2 RNA. The workflow of this assay is a fast targeted NGS reporting the results in 1 day, which is capable of detecting 20 viral copies in the sample [161].

Wang et al. have developed another nanopore target sequencing (NTS) for the detection of respiratory viruses including SARS-CoV-2 simultaneously in 6–10 h. The LoD of this assay is 10 copies/mL with the capability of detecting more than one infection simultaneously. Some of the merits of this test are lower cost in comparison with whole-genome sequencing, rapid TaT in the same day, a wide range of detection, and a lower rate of false-negative compared with RT-PCR. On the other hand, this test is limited due to failure in the detection of the nucleic acid fragments 300–950 bp in length, which significantly decreases the sensitivity of the test. Additionally, although the test is faster compared to the other sequencing technologies, it is still longer than qPCR or other PoC tests that are only acceptable for lab-based purposes. The throughput of this system is low, and the process includes opening the lid of the tubes, increasing the probability of contamination [162]. Some other companies have developed sequencing-based tests for COVID-19 studies (Table 4).

BillionToOne (Menlo Park, CA, USA) is a cancer molecular diagnostics company that has developed the Sanger sequencing-based molecular diagnostic tool called qSanger-COVID-19 test using qSanger. This FDA EUA approved test is very similar to the traditional Sanger sequencing and significantly faster than PCR due to the throughput of 1536 samples on qSanger. The workflow of qSanger COVID-19 assay includes reverse transcription and RT-PCR amplification of both SARS-CoV-2 target sequences and synthetic spike-in DNA in the master mix followed by Sanger sequencing the products. The data obtained from the resulted chromatogram are used to report a positive or negative result for the COVID-19 suspected specimen [163]. Performing the test requires the presence of expert sanger sequencers in the laboratory [164].

Due to the nature of sequencing, this methodology is not suitable for PoC and fast detection of COVID-19 in most cases; mNGS is restricted by many factors, including turnover time, high probability of contamination, requiring highly trained technicians and high costs, which is the main barrier to introduce this technology as a diagnostic method [78]. Another limitation associated with such non-propagative diagnostic laboratory works is their mandatory conduction using biosafety level 2 (BSL-2) containment procedures and facilities. Performing NGS requires specific equipment and multiple manual interventions. Future simplification and automation may turn this strategy into a routine diagnostic plan by reducing the HoT, the potential contamination, and human errors [165]. However, by emerging a portable and real-time sequencing device by Oxford Nanopore (Oxford, UK), particularly MinION and Flongle adapter, this technology offers a way forward for PoC diagnostics and brings sequencing to the near patient setting. Oxford Nanopore company has developed rapid sequencing tests for the detection of SARS-CoV-2 [166]. A nanopore sequencer can be used for sequencing the DNA or RNA molecules relying on the conversion of the electrical signal of the nucleotides, which pass through a nanopore [167]. They have announced their LamPORE assay as a low-cost and rapid test for COVID-19 detection. This test is based on real-time nanopore sequencing technology combined with amplification of the viral RNA in the original sample for library preparation using LAMP, targeting E, N ORF1a, and a control gene. It analyzes a large number of samples simultaneously with a new barcoding approach, and the results of 12 extracted RNA samples are reported in one hour [168,169]. The ARTIC network has also proposed a protocol for rapid nanopore whole-genome sequencing of SARS-CoV-2 with a reduced reagents cost, decreased HoT, and employing 22 additional primers to achieve more coverage using the portable Oxford Nanopore MinION sequencer, which is smaller than a cellphone. This is the new version of their previous published protocol for SARS-CoV-2 genome sequencing, which became the most popular strategy worldwide due to being cost-effective and simple [170]. This strategy requires 7 h for the steps of reverse transcription, PCR, adding barcode, adding adapter, sequencing, and analysis. This methodology is based on direct detection of the target virus via tiled, multiplex primers with high sensitivity with the advantage of using clinical samples directly as input compared with metagenomic approaches [171,172].

**Table 4 bioengineering-08-00049-t004:** A selected list of the sequencing-based targeting SARS-CoV-2.

Manufacture	Test	Test Format	Target	Time to Result	LoD	Sensitivity	Specificity	Regulatory Status
IDbyDNA (Salt Lake City, UT, USA) [173]	NGS-Based SARS-CoV-2 Detection test	NGS-based metagenomics	(No info)	(No info)	(No info)	(No info)	(No info)	Used in Indonesia
BGI Genomics (Beijing, China) [174]	DNBSEQ-T7 2019-nCoV	Combination of RT-PCR and meta- genomics detection (combinatorial probe anchor synthesis sequencing)	(No info)	Results in a few hours	(No info)	(No info)	(No info)	RUO
Helix OpCo, LLC (San Mateo, CA, USA) [175]	Helix COVID-19 NGS Test	NGS	S gene	2–4 h	125 genomic copy equivalents/mL	100.0%	100.0%	FDA EUA
BillionToOne [176]	qSanger-COVID-19 Test	Sanger Sequencing Combining the Sanger sequencing and the machine learning algorithm	N protein	(No info)	3200 copies/mL	(No info)	No cross-reaction is expected	FDA EUA
YouSeq (Hampshire, UK) [177]	SARS-COV-2 Coronavirus NGS Library Prep Kit	Complete kit for amplicon-based NGS Library preparation, Amplicon-based protocol	99.5% viral genome coverage	≈9 h	(No info)	(No info)	(No info)	RUO
Illumina Inc. [160]	Illumina COVIDSeq Test	NGS High-throughput Shotgun metagenomic sequencing	Detects 98 targets on SARS-CoV-2	1536 to 3072 results can be processed in 12 h	(No info)	1000 copies/mL	(No info)	FDA EUA
Oxford Nanopore [178]	LamPORE COVID-19	NGS High-throughput-combines nanopore analyses with loop-mediated isothermal amplification	SARS-CoV-2 genes including E, N ORF1a	Under two hours	7–10 copies/µl	98%	100%	CE-marked
Oxford Nanopore [178,179]	LamPORE	NGS High-throughput nanopore sequencing	Entire viral genome (>99%) using the ARTIC network	Provide a consensus viral genome in 7 h	(No info)	(No info)	(No info)	RUO
Thermo Fisher [180]	Ion AmpliSeq SARS-CoV-2 Research Pane	Targeted sequencing by overlapping amplicons	Entire viral genome	14 h	20 copies	(No info)	(No info)	RUO
Paragon Genomics Inc. (Hayward, CA, USA) [181]	CleanPlex SARS-CoV-2 Research and Surveillance NGS Panel	NGS Highly multiplexed amplicon-based target enrichment	Entire viral genome except for 92 bases at the ends using 343 primer pairs	5.5 h with Less than 1 h HoT	3.9 copies/reaction for the E gene assay and 3.6 copies/reaction for the RdRp assay	E gene and RdRp gene assays (5.2 and 3.8 copies per reaction respectively)	(No info)	RUO
Fulgent Genetics/MedScan Laboratory (Williston, ND, USA) [182]	COVID-19	NGS	(No info)	2–4 days	(No info)	(No info)	(No info)	FDA EUA
Guardant Health (Redwood City, CA, USA) [183]	Guardant-19	Reverse Transcriptase PCR (RT-PCR) and NGS	N1 region of the SARS-CoV-2 N gene and human RNase P gene	2–4 h	125 copies/mL	95%	98%	FDA EUA
Twist Bioscience (San Francisco, CA, USA) [184]	NGS-based target capture for SARS-CoV-2 detection and screening	NGS-based target capture	Entire viral genome	(No info)	10 copies	Coverage of >99.9% of the genome	(No info)	RUO
Clear Labs, Inc. (San Carlos, CA, USA) [185]	Clear Dx SARS-CoV-2 Test	Automated (manual RNA extraction) and high-throughput (192), Multiplexed barcoded RT-PCR and targeted NGS	21 target genes	2–4 h	2000 copies/mL	100%	Cross-reaction with SARS-CoV-1 in one sequence	FDA EUA
University of California, Los Angeles (UCLA) (Los Angeles, CA, USA) [186]	UCLA SwabSeq COVID-19 Diagnostic Platform	NGS High-throughput RT-PCR and Sequencing	S2 gene	12 h	250 genome copy equivalents/mL	100%	Not expected	FDA EUA

## 4. Emerging Detection and Sensing Strategies

The current benchtop strategies require multiple steps of human intervention and set-up process. Due to the increasing demand for rapid, cost-effective, less complex and accurate tests for COVID-19 detection, novel technologies have attracted researchers’ attention to make up for the shortcomings in a testing area [187]. One of the solutions can be employing lab-on-a-chip (LoC) strategies to perform all of the tests on one device. Such microfluidic-based biosensors with glass, polymeric, paper-like, or silicon substrate can be used for on-site detection by acting as miniaturized laboratories and have the potential to be adapted for targeting NA from various pathogens [188].

Among biosensing technologies, field-effect transistor-based biosensors (FET) have demonstrated to be promising strategies based on their miniaturized size, fast and sensitive response, and parallel sensing with the potential of being used in PoC setting [189]. A cleavage-based approach has been developed for sensing which combines powerful graphene FET (GFET) with a sensitive CRISPR/Cas system by immobilizing CRISPR/Cas on graphene field effective transistors (gFET) to target specific sequences in a CRISPR-Chip [190]. In one of the proposed tests with this technology, the catalytically deactivated Cas9 (dCas9) CRISPR complex (dRNP) functionalizes the graphene and interacts with the target sequence by scanning the genetic material [189]. Liquid-gate electrodes are in direct contact with the mixture of reaction buffer and the sample constantly. The current between two source and drain electrodes of the graphene channel is controlled by the applied voltage between the source electrodes and liquid-gate. Immobilized dRNP hybridized with negative-DNA alters the conductivity of the channel and counter accumulates to generate a stable neutral charge, which produces an ion-permeable layer on the surface (Figure 3) [191]. Different ion concentrations between this permeable layer and bulk solution creates Donnan potential, which changes the electrical field between gate electrodes and the source, and the final result is the ability to sense the DNA [192].

Smartphones have recently been integrated with microfluidic biosensing technologies for optical detection and analysis of the signals. These smartphone-assisted sensors are promising devices to be used for the detection of the infection in the early stages of the disease [193]. Another portable device that can improve the traditional PCR tests in using dual heating elements for amplification of nucleic acid. Designing the devices with built-in dual heaters yields the significant advantage of reduced cost and size of the instrument [194]. Such all-in-one microfluidic PCR systems can be employed for on-site detection of the pathogens in a PoC setting [195].

Jing Wang et al. have introduced a sensor-based optical detection test. They suggested a dual-functional plasmonic biosensor combining plasmonic photothermal (PPT) heating effect and localized surface plasmon resonance (LSPR) sensing transduction as a promising solution for COVID-19 detection. In this technology, two-dimensional gold nanoislands (AuNIs) are functionalized with thiol-cDNA (RdRp-COVID-C) ligands as reporters DNA and target sequences from SARS-CoV-2 genetic material. Two-dimensional (2D) AuNIs successfully generated local PPT heat and transduced the hybridization (Figure 4). The results were high sensitivity and accuracy of the dual-functional LSPR biosensor allowing specific detection of SARS-CoV-2 with LoD down to the concentration of 0.22 pM. This diagnostic platform is proposed as a potential clinical test besides PCR-based diagnostics [196].

Luminostics Inc. is a company developing PoC tests. In their products, they utilize reporters based on Luminostics’ unique technology using smartphone’s optic, an iOS/Android application, and an affordable reusable adapter [197,198]. They have also developed their technology for the detection of respiratory infectious pathogens. In this PoC NA-based test, LAMP is used for the generation of 109 copies in less than 60 min and a readout instrument for end-point fluorescent detection of the emission from a EvaGreen DNA intercalating dye in the microfluidic chip RT-LAMP assay. This test could detect the virus from the nasal swabs’ media. The current device detects five respiratory pathogens and could be referred as a model system for infectious diseases such as COVID-19. Briefly, on-chip detection initiates after insertion of the controls, targets primers, and LAMP reaction mix followed by heating at 65 °C. Then, the chip is inserted in the cradle for imaging, and the results are reported in 30 min (Figure 5). The sensitivity of this inexpensive portable test for the early detection of EHV1 was reported to be 5.5 × 10^4^ copies/mL, corresponding to about 18 copies per reaction, which is adequate and comparable with PCR-based assays [199].

Taken together, the current methods such as sequencing and PCR are time-consuming, and there is always a probability to provide false results. Additionally, they are incapable of fulfilling the current challenges and demands for more accurate and direct PoC detection of the pathogens in the current and future pandemics. There are many emerging strategies in this category including aptamer-based bio-navigate, electrochemical and optical biosensor, DNA hydrogel formation by isothermal amplification of complementary target (DhITACT-TR) chip-based, surface plasmon resonance (SPR), junction-gate field-effect transistor (JFET), or metal–oxide–semiconductor field-effect transistor (MOSFET), graphene-FET, Ag/Au-based electrochemical biosensor, and surface plasmon platforms, which have recently gained the attraction of the researchers to hopefully overcome the shortcomings, develop more promising detection kits, and pave the way to more rapidly controlling the viral spreads of future hazards [200]. These novel technologies have demonstrated success in detecting the pathogens such as SARS or MERS viruses previously, and they are readily available for mass production as cost-effective and miniaturized detecting devices with very low detection limits and high sensitivities, which could be either combined with the current strategies or developed for on-sight detection of the pathogens from sample preparation to signal detection [201]. These integrated microsystems have an auspicious future in the early detection of diseases, particularly viral pandemics such as the current global health burden.

## 5. Discussion

In this review, we discussed the current techniques and promising alternatives which have been developed or are in development for COVID-19 detection, or they have the potential to be setup to control and harness any probable viral spread during the pandemic outbreaks. Although NA-based tests are the gold standard and reliable for early detection, their applicability is limited due to many challenges. Evidence demonstrates that rapid evolution and genetic diversity have been affecting SARS-CoV-2 genotype from the initiation of the pandemic, producing variations that may be located in the sequences complementary to the designed primers and probes in NA-based tests [158,202]. Hence, the targets should be precisely designed to minimize the probability of occurring mismatches, increase the sensitivity and specificity of the test, and reduce false results. The presence of mismatches between the primers and their target ends in a reduction in reporting false-negative results, especially in the mutant and novel variants of coronavirus 19. Some of the current kits may not be able to detect the mutant viruses such as the U.K. variant in the samples [202,203,204]. Hence, it is critical to recognize these evolutionary hotspots and avoid targeting them when designing the test. The optimum primer designs include more than one gene on virus RNA targeting, at least one conserved or species-specific sequence along with at least one SARS-CoV-2-specific sequence to increase the accuracy of the test [155,205].

Additionally, employing high-quality primers and probes directly increases the accuracy of the RT-PCR amplification [158]. Contamination is another issue that may occur during any steps of the test, resulting in a false positive. It should be highlighted that although some manufactures claim their test to be 100% accurate, none of the developed test’s assays are capable of achieving specificity and sensitivity of 100% owing to a variety of limitations including errors caused during sample collection, sample transportation, using reagents lacking the standards, and technical or executive faults [206].

The lesson is learned from COVID-19, and we should be prepared to face future viral attacks, since it takes at least a few months to develop a functioning vaccine for a novel pathogenic disease. Hence, vaccination is not the sole solution, and it is critical to standardize diagnostic technologies to be adapted for targeting new microorganisms rapidly and reliably during the early months of their spread when any vaccines have not received approval yet. To overcome the current challenges of NA-based tests including the complexity, high costs, long durations, requiring trained laboratory personnel, cross-contamination, false results, shortage of the required materials, and consumptions and more importantly lack of reliable miniaturized and fully automated at-home tests for self-assessment or disease follow-up, researchers have proposed alternatives to more promising state-of-the-art methodologies such as electronic biosensors with excellent accuracy for monitoring human cells as well as pathogens. Innovations in modern medical technology are highly desired to enable the rapid selection of effective drugs for the treatment of infectious diseases. For most infectious diseases, early and effective treatment is crucial to avoid costly and perhaps even lethal complications. Sensitive and scalable PoC tests would increase the scope for COVID-19 diagnosis to be made in the community and outside the laboratory setting. The development and mass production of cost-effective, easy-to-use, accurate, and fast PoC and point-of-need (PoN) diagnostic devices would potentially eliminate the current workload and labor for the technicians, fewer hospitals and health care providers will be engaged with the disease, and the doctors will be free to take care of the patients with other crucial requirements than COVID-19.

As mentioned before, due to obtaining unsatisfactory results from the current detection tools, the variety of alternative sensors might be useful for prompt control of the viral spread in the future pandemics in the integrated fully automated systems. The authors would suggest developing novel low-cost micro fabricated fully automated devices for sample preparation and detection. The mass production of such technologies can lower the costs and make it affordable for many people. To date, many efforts have been made to miniaturize the lab instruments used for molecular analysis using standard technologies such as complementary metal-oxide semiconductors (CMOS). CMOS technology is a useful integration strategy and can lead to integration of the biosensors and microfluidic systems in one single chip in order to develop of portable low-cost PoC devices in the urgent pandemic situations and being economically produced in series right into the market. CMOS technology has proven to be useful in detection of the pathogens. There are various types of CMOS-based electrochemical biosensors including potentiometric [207,208], voltammetric [209], impedimetric [210], and capacitive [211] sensors that are useful for the NA-based diagnosis of infectious diseases. In one research, Malpartida-Cardenas et al. coupled 64 × 64 arrayed electrochemical ion-selective field-effect transistors (ISFETs) fabricated in unmodified CMOS technology with LAMP for *Plasmodium falciparum* malaria diagnosis and artemisinin-resistance detection [207]. Hsu et al. have also reported an electrolyte–insulator–semiconductor (EIS) sensor array using a polar-mode measurement method for the detection of Zika Virus oligonucleotides [212]. Hence, CMOS has demonstrated great advantages for biosensing DNA and can be adopted for COVID-19 or the other pandemics. Additionally, microfluidic technologies have been greatly grown in the fold of bioengineering and proposed many solutions for DNA and RNA sample preparation [213,214]. These advances also have the potential to be adopted for COVID-19 diagnostics. There are many other microfabrication technologies that can be adequately selected and adopted for infectious detection purposes using the molecular technique. These methods can be standardized for low-cost PoC purposes after receiving FDA.

Taken together, more promising PoC and PoN diagnostic tests should be employed to overcome the limitations and shortcomings of the current strategies. Developing such biosensors will ease the way of real-time monitoring of the cells with cost-effective, rapid, and accurate home-use tools shortly to be prepared for future pandemics. This review paper can help researchers further evolve the biosensors for limiting the growth of life-threatening pandemics.

## Figures and Tables

**Figure 1 bioengineering-08-00049-f001:**
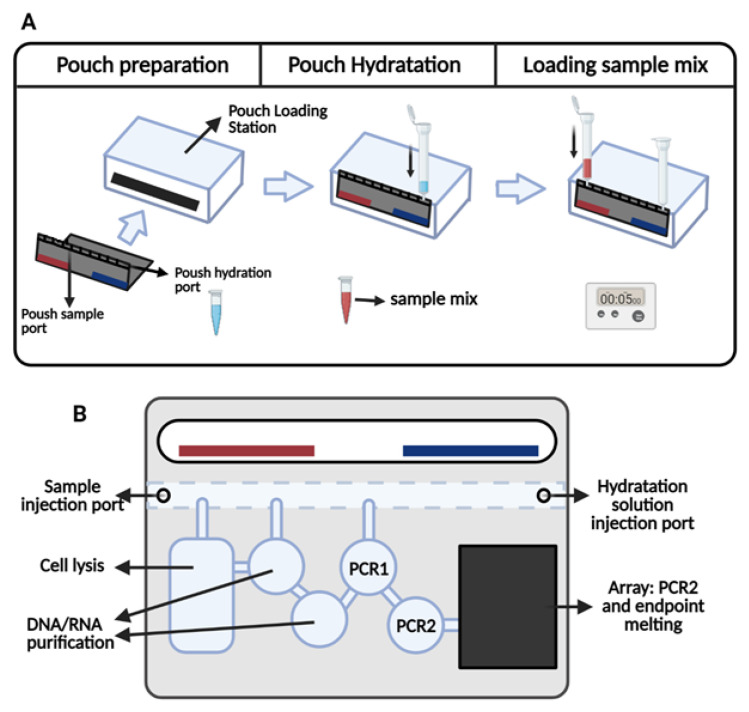
(**A**) Schematic view of the BioFire^®^ COVID-19 Test quick workflow including the insertion of the pouch into Pouch Loading Station with sample vial and hydration injection vial, pushing down to puncture seal and hydrating the solution, adding specimen to the sample injection vial, loading sample mix into the pouch in 5 s, discarding the vials, followed by inserting the pouch into the system and running the test. (**B**) Schematic view of FilmArray^®^ system. After sample injection, the system lyses the sample by agitation, extracts and then purifies NA using magnetic bead technology and Performs nested multiplex PCR: PCR1 a reverse transcription multiplexed reaction, PCR2 multiple singleplex second-stage PCR reactions amplifying PCR1 products.

**Figure 2 bioengineering-08-00049-f002:**
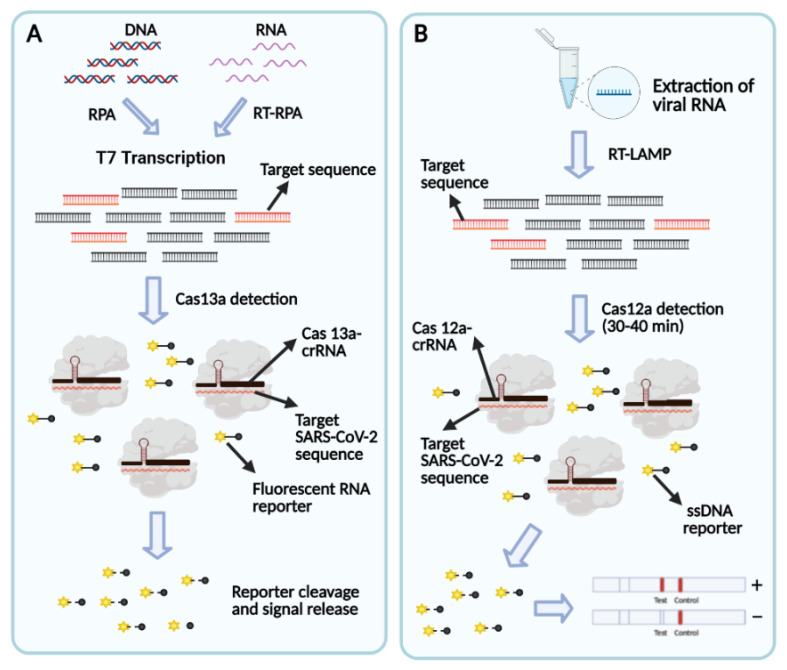
(**A**) Schematic of SHERLOCK detection assay. A pre-amplification step is started with RNA or DNA sample inputs, the amplicons are converted to RNA using T7 transcription and then detected using Cas13−crRNA complexes. This detection is followed by cleaving and activating the fluorescent RNA reporters. (**B**) Schematic for SARS-CoV-2 detection using SARS-CoV-2 DETECTR. RNA extraction is followed by an RT-LAMP pre-amplification step and a subsequent Cas12-based detection, which is visualized by an LF reader.

**Figure 3 bioengineering-08-00049-f003:**
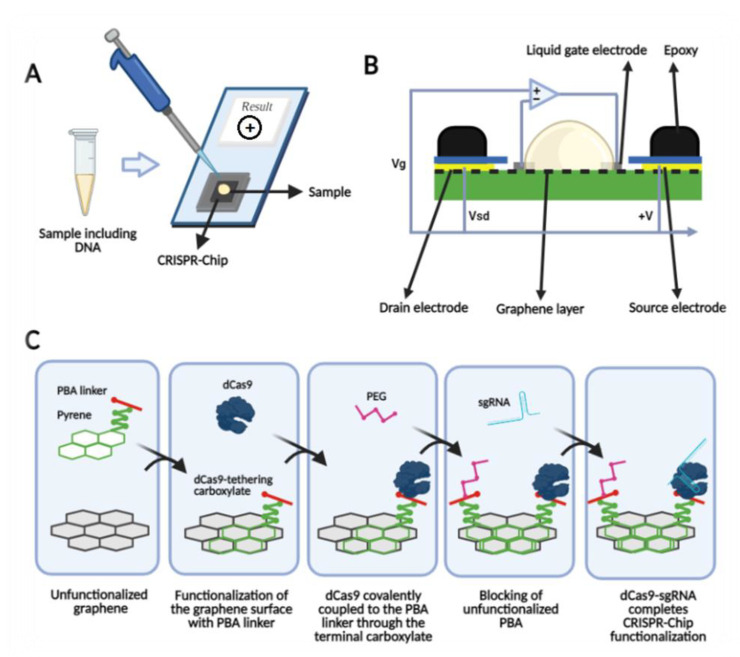
The schematic view of CRISPR–Chip functionalized with CRISPR–dCas9. (**A**) the CRISPR–Chip reports the results in 15 min; (**B**) the process of functionalization of the graphene surface; and (**C**) the components of the CRISPR–Chip including a liquid gate directly in contact with the sample, and three terminal gFETs utilizing functionalized graphene between source and drain electrodes.

**Figure 4 bioengineering-08-00049-f004:**
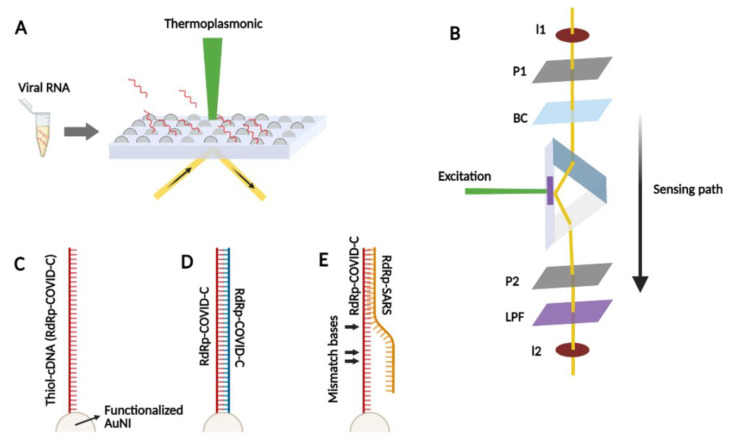
The schematics of the developed dual-functional plasmonic biosensor. (**A**) the overall structure of the sensor combining the PPT effect and LSPR for detection of the virus genetic material. (**B**) The dual-functional PPT enhanced LSPR biosensing system including the aperture-iris (I1/I2), the linear polarizers (P1/P2), the birefringent crystal (BC), and totally reflected at the interface of AuNI-dielectric for LSPR detection. A laser diode (LD) generates the PPT effect on AuNIs. (**C**) AuNI functionalization based on the reaction with thiol-cDNA ligands. (**D**) Illustration of the hybridization between two complementary strands, the reporter DNA, and the target sequence in SARS-CoV-2 genome. (**E**) Specific hybridization of the functionalized thiol-cDNA and inhibiting partial adhesion of RdRp-SARS sequence with two mismatches.

**Figure 5 bioengineering-08-00049-f005:**
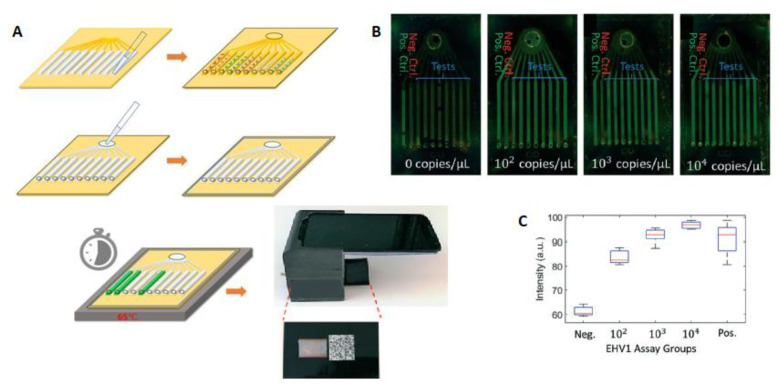
The workflow of the LAMP-based on-chip detection, (**A**) Deposition of the primers and controls, injection of LAMP reaction mix, heating the chip at 65 °C and insertion into the cradle for end-point fluorescence imaging for 30 min. (**B**) different concentrations of EHV1 templates were amplified on the chips (from left to right, channel 1: positive control, channel 2: negative control and channel 3–10: EHV1 primers). (**C**) the average intensity reported for the channels for each assay [199].

## Data Availability

Data is contained within the article.

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
