# Peer review of "COVID-19 Diagnostic Strategies. Part I: Nucleic Acid-Based Technologies"

_bioengineering, 2021, doi:10.3390/bioengineering8040049_

Round 1
Reviewer 1 Report
This manuscript summarizes most important molecular diagnostic methods for COVID19. I think it is qualified to publish after some modifications. Here are my suggestions:
- Table 1, (i) the title can be replaced with “A selected list of the ““real time PCR””-based tests and their performance for COVID-19 detection”. (ii) The column of “Technology” can be deleted. (iii) Separate the column “Sensitivity” and “LoD”.
- Please explain the LoD was 625 copies/mL in Line 188 and 2 copies/mL in Line 195.
- Line 209, the sentence “The advantages of these kits are increased sensitivity, independence to the real-time 209 thermal cyclers, being cost-saving and the wide availability of the reagents compared with 210 RT-PCR [48]” is not applicable to all the nPCR methods. I suggest the authors delete it.
- Line 248 and 264, these two paragraphs discuss the limitation of PCR but are not directly related to nRCR, so they are not suitable to be here. Since the mentioned problems exist in all NA testing, I suggest move these two paragraphs to the Dicussion part.
- Table 2, (i) the title can be replaced with “A selected list of the LAMP-based tests and their performance for COVID-19 detection”. (ii) The column of “Technology” can be deleted.
- Line 342, TMA (Transcription-Mediated Amplification) à Transcription-Mediated Amplification (TMA)
- Line 346, delete “patented”. (As most methods are patented)
- Line 371-374, it is not a proper comment. Because CRISPR/Cas system is only for signal generation and need to be coupled with other amplification methods, it would not be “rapid, simple, inexpensive, etc”, compared with other methods, which the authors have mentioned in Line 381.
- Table 3, (i) Simplify the columns “Technique” and “Detection”. (ii) Why does this table have a column “Regulatory status”? it does not appear in other tables.
- Line 476, please revise this sentence.
Line 507, the BillionToOne developed qSanger method is not an NGS-based method, so it is not suitable to be in this paragraph.
- Figure 3, this figure is not important. Besides, the description is not clear.
- Line 543, is this HoT or ToT?
- Table 4, there are two Sensitivity columns.
- Line 606, the comments of this paragraph should be revised, as most of these biosensors have not yet been proved to be useful in the market.
- The discussion part emphasizes too much about biosensors. Do the authors think biosensors is the best answer of NA test in the future?
- Need provide full name for some abbreviations: HoT, LAMP, LFA. Also please check the case of abbreviations.
- Add references for all tables.
Author Response
Dear Reviewer,
Thank you very much for your time and for reviewing this manuscript. You have mentioned very critical points which helped us to improve our manuscript. We tried to apply your valuable comments in the revised version in the best way.
Sincerely,
Tina Shaffaf, Ebrahim Ghafar-Zadeh

Reviewer 2 Report
In this manuscript, the authors reviewed nucleic acid-based technologies for COVID-19 diagnostic testing. The authors covered PCR-based tests, isothermal nucleic acid amplification-based tests, sequencing-based tests, and biosensors. For each category of tests, the authors provided fairly thorough discussion including the advantages and the disadvantages. The authors also framed their discussion around the need for point-of-care detection.
This manuscript provides a solid contribution to the rapidly expanding list of nucleic acid-based technologies for COVID-19 diagnostic testing. The reviewer agrees with the authors that the list of technologies that they reviewed was comprehensive, thereby providing a useful review for the community. Nevertheless, the current manuscript can and should be improved. For example, the reviewer thinks that the prospect of biosensors should be justified more strongly, and the introduction of point-of-care use should be made more systematically. The current manuscript is also a little thin on the displayed items, so the reviewer has some suggestions for adding more figures and tables. Additional major and minor comments are as followed:
Major Comments
- The reviewer disagrees with the authors that biosensors would provide useful solutions that can supplant nucleic acid-based technologies for detecting SARS-CoV-2 or pathogens that may cause pandemics in the future. This is because biosensors typically have insufficient analytical sensitivities. Using two references listed by the authors in this manuscript, Reference 156 reported a limit of detection of 0.22 pM and Reference 151 reported a limit of detection of 1.7 fM. Both are orders of magnitude worse than what nucleic acid-based technologies can achieve. Also, biosensors that target nucleic acids still require sample preparation and detection instrumentation, just like nucleic acid-based technologies. The reviewer asks to either provide better examples to support this claim or modify the claim accordingly.
- Nevertheless, the reviewer found Reference 151 interesting and showed enough potential for point-of-care use. As the manuscript currently is a little thin on the figures, the authors could consider adding the schematic from Reference 151 and perhaps a few other works as a new figure, which can better illustrate the promise of biosensors.
- The reviewer agrees with the authors in that bringing nucleic acid-based technologies to the point of care would maximize their impacts for combating transmissible infectious diseases that can cause pandemics in the future. However, the reviewer believes that the authors should address this need more systematically. Specifically, the authors should dedicate at least a paragraph to discuss the need for point-of-care of diagnostics and the requirements for nucleic acid-based technologies for achieving point-of-care use – such as sample processing and platform miniaturization and automation. The sequencing section does not fit snuggly with point-of-care use, but with the advances made by Oxford Nanopore (e.g., MinION and Flongle), there are certainly efforts made toward bringing sequencing to the point of care. So the authors could also add a few more sentences to their sequencing section.
- The authors mentioned in the conclusions (lines 621 to 624) that “it is critical to standardize diagnostic technologies to be adapted for targeting new microorganisms rapidly and reliably during the early months of their spread when any vaccines have not received approval yet.” The reviewer agrees with the authors that this would be ultimately the goal. But the reviewer was disappointed to find no suggestions from the authors on how to do that. During this pandemic, the “standards” from the United States seem to be set by the FDA through EUA. Consequently, although there are numerous variations in diagnostic testing, they at least meet certain criteria outlined by a centralized authority. The reviewers would like to know if the authors have other suggestions that can accelerate or improve the process. The reviewer thus asks the authors to critical address this claim and make the necessary changes and improvements to their manuscript. Such an edit could also serve to improve the discussion section of the manuscript, which is currently a little short.
- The reviewer only had time to look through a few references in more detail but already found some mistakes. For example, the authors incorrectly stated that Reference 77 reported 8 primer sets. In Reference 77, the ORF1ab assay used 5 primers and the S gene assay used 6 primers, and the two assays were performed in separate reaction tubes. In another example on lines 589 to 590, the authors cited Reference 153 to support the statement “These smartphone-assisted sensors can be used for the detection of the infection in the early stages of the disease.” But Reference 153 describes a phone-based detector of lateral flow strips that detects DNA in the nanomolar range. Reference 153 therefore cannot provide adequate support for the claim that the authors made. As a tangent, the reviewer also questions why Reference 153 was selected in the first place. The limit of detection reported in Reference 153 means that nucleic acid amplification would have been needed prior to detection using the reported detector, but because end-point detection of nucleic acid amplification usually cannot be truly quantitative, the reviewer cannot find a meaningful advantage of this detector over qualitative detection using naked eye. The reviewer asks the authors to carefully go through the references one more time to ensure accuracy.
- On lines 609 – 614, the authors listed many of biosensing techniques without discussing why they consider those as promising techniques. To include those techniques, the authors should use examples to discuss what these biosensors was able to achieve (limit of detection, degree of miniaturization and instrument integration, etc.), which would provide a stronger rationale for why they are promising. The authors should also avoid directly using abbreviations of these techniques.
- On line 625, the authors mentioned that “to overcome current challenges”, which referred to the challenges that existing nucleic acid-based technologies face. As these challenges were scattered throughout the manuscript, the reviewer thinks it would be helpful to provide a short summary or even make a table so that these challenges so that potential readers can more conveniently take these important points home.
Minor Comments
- On lines 47 – 49, the authors stated: “Such standardized and reliable technologies will protect thousands of lives especially in the first months of the future pandemics when no vaccines are present [6].” The reviewer found the placement of Reference 6 misleading because it is about vaccines rather than diagnostic technologies. The reviewer would like the authors to find more appropriate references for supporting that reliable technologies will protect lives.
- On line 311, the authors mentioned that “LAMP targets multiple sequences.” The reviewer would like the authors to clarify what they mean by this. Multiplex detection is possible for LAMP, albeit not an inherent advantage of LAMP. Did the authors refer LAMP primers target four to six regions within the target?
- On lines 633 and 637, the authors twice used the term “point-of-need” and provided two different abbreviations. The reviewer asks the authors to correct the minor inconsistencies.
- The authors also need to properly define terminologies before using abbreviations. Some examples in the manuscript include EUA, HoT, TaT, and ToT. The last one may be a typographical error. The authors are asked to correct it if that is the case.
Author Response

(The authors gave the same response as above.)

Round 2
Reviewer 2 Report
The revised manuscript addresses the reviewer’s comments and indeed improves upon the original manuscript. The reviewer has a small suggestion for Section 4. As before, the reviewer generally disagrees with the viability of “biosensors” as nucleic acid-based diagnostic tools. With its current content, the reviewer thinks that the authors could simply change the section heading to “Emerging Detection and Sensing Modalities” (or something similar), which still gets the central messages across.
Author Response
Dear Reviewer,
Thank you for your time and for suggesting a more appropriate heading for Section 4. The related heading was changed to "Emerging Detection and Sensing Strategies".
Sincerely,
Tina Shaffaf. Ebrahim Ghafar-Zadeh